# What General Neurologists Should Know about Autoinflammatory Syndromes?

**DOI:** 10.3390/brainsci13091351

**Published:** 2023-09-21

**Authors:** Marianna Pinheiro Moraes de Moraes, Renan Rodrigues Neves Ribeiro do Nascimento, Fabiano Ferreira Abrantes, José Luiz Pedroso, Sandro Félix Perazzio, Orlando Graziani Povoas Barsottini

**Affiliations:** 1Department of Neurology, Universidade Federal de São Paulo, São Paulo 04039-002, Brazil; marianna.m.moraes@gmail.com (M.P.M.d.M.); fabianofabrantes@gmail.com (F.F.A.); zeluizpedroso@yahoo.com.br (J.L.P.); 2Departament of Rheumatology, Universidade Federal de São Paulo, São Paulo 04039-050, Brazil; renanrnrnascimento@gmail.com (R.R.N.R.d.N.); sperazzio@yahoo.com.br (S.F.P.)

**Keywords:** autoinflammatory disease, hereditary periodic fever syndromes, hereditary recurrent fever, hereditary autoinflammatory disease

## Abstract

Autoinflammatory disorders encompass a wide range of conditions with systemic and neurological symptoms, which can be acquired or inherited. These diseases are characterized by an abnormal response of the innate immune system, leading to an excessive inflammatory reaction. On the other hand, autoimmune diseases result from dysregulation of the adaptive immune response. Disease flares are characterized by systemic inflammation affecting the skin, muscles, joints, serosa, and eyes, accompanied by unexplained fever and elevated acute phase reactants. Autoinflammatory syndromes can present with various neurological manifestations, such as aseptic meningitis, meningoencephalitis, sensorineural hearing loss, and others. Early recognition of these manifestations by general neurologists can have a significant impact on the prognosis of patients. Timely and targeted therapy can prevent long-term disability by reducing chronic inflammation. This review provides an overview of recently reported neuroinflammatory phenotypes, with a specific focus on genetic factors, clinical manifestations, and treatment options. General neurologists should have a good understanding of these important diseases.

## 1. Introduction

Inflammation is an essential physiological response of all metazoan organisms to harmful agents. Unlike autoimmune diseases, which are mediated by the adaptive immune system, the term “autoinflammation” refers specifically to inflammatory processes mediated by the innate immune system [1].

The concept of systemic autoinflammatory diseases (SAIDs) was proposed by McDermott et al. in 1999 to describe a group of disorders resulting primarily from dysregulation of the innate immunity and hypersecretion of pro-inflammatory cytokines, with minimal to no T or B cell involvement [2]. These diseases are caused by abnormal expression of proteins considered essential for the homeostasis of the innate immune system and are secondary to inherited or de novo mutations [3].

Systemic autoinflammatory diseases usually present as recurrent episodes of non-infectious fever and multisystem inflammation, primarily affecting the serous and synovial membranes, skin, and eyes. Myositis, vasculitis, and systemic amyloidosis may also occur [4]. Severe recurrent inflammatory attacks are the hallmark of SAIDs. Typical clinical signs include malaise, fever, skin rash, arthritis/arthralgia, abdominal pain, and elevated levels of acute phase reactants. Patients are usually asymptomatic during the remission phase. However, subclinical inflammation may occur [4].

Hereditary recurrent fevers (HRFs) are a well-known subgroup of SAIDs. Several HRFs have been described, including familial Mediterranean fever (FMF), tumor necrosis factor receptor-associated periodic syndrome (TRAPS), cryopyrin-associated periodic fever syndrome (CAPS), and hyperimmunoglobulin-D with periodic fever syndrome (HIDS—or mevalonate kinase deficiency, MKD). Over time, the concept of autoinflammation has been expanded to encompass other Mendelian and polygenic diseases, such as deficiency of adenosine deaminase-2 (DADA-2), monogenic interferonopathies, Behçet disease, and Still disease [4].

Although uncommon, neurologic symptoms are being increasingly recognized as part of the clinical spectrum of SAIDs. In this review, clinical and immunopathological concepts are discussed, with particular emphasis on neurological manifestations.

## 2. Pathophysiological Mechanisms

Acquired autoimmune diseases like systemic lupus erythematosus (SLE), rheumatoid arthritis (RA), and ANCA (antineutrophil cytoplasmic antibodies)—associated vasculitis are usually associated with immune tolerance breakdown and organ damage induced by auto-reactive T cells and autoantibodies [5]. In contrast, SAIDs result from innate immunity dysfunction and consequent hyperactivation of phagocytes. Autoantibodies and autoreactive T-cell clones are typically absent (Figure 1) [6,7].

A deeper understanding of the pathophysiology of SAIDs provided additional valuable insights into innate immune system activation and proper function [8,10]. The initiation of innate immunity can be triggered by the activation of Pattern Recognition Receptors (PRRs). These receptors act as sensors for microbe-specific molecular signatures (known as pathogen-associated molecular patterns, or PAMPs) and self-derived molecules from damaged cells (known as damage-associated molecular patterns, or DAMPs), and include members of several receptor families, such as extracellular Toll-like (TLRs), nucleotide-binding oligomerization domain (NOD)-like (NLR), retinoic acid-inducible gene-I-like, C-type lectin and absent in melanoma 2 (AIM2)-like [9,11].

Sensing by PRRs activates specific pro-inflammatory intracellular pathways, leading to the expression of multiple cytokines, chemokine, and antibiotic-resistance genes. For example, NLR activation triggers the formation of multiprotein complexes (the so-called inflammasomes), which in turn promote the oligomerization and activation of inflammatory caspases involved in IL-1β and IL-18 cleavage and activation (such as caspase-1) and induce an inflammatory form of cell death known as pyroptosis [12]. These pathways are precisely regulated by specific proteins at multiple levels. Impairment of regulatory mechanisms by loss-of-function (LOF) mutations or hyperactivation of effector components by gain-of-function (GOF) gene variants create a chronic inflammatory microenvironment driven by constitutive activation of a molecular pathway, resulting in tissue damage [11].

Nuclear factor kappa-light-chain-enhancer of activated B cells (NFkB) is a family of transcription factors implicated in apoptosis, tumorigenesis, inflammation, and autoimmune diseases. In resting cells, NFkB is inhibited by IkB (Inhibitors of NF-kB). Upon appropriate stimulation, it is translocated to the nucleus and drives the transcription of key pro-inflammatory genes, such as IL1-β and TNF [13]. Pathogenic mutations affecting this network disrupt normal NFkB function, causing immunodeficiency and NFkB-related autoinflammatory diseases (or relopathies, since NFkB proteins belong to the REL family of proteins) (Figure 2) [14].

## 3. Classification of SAIDs

Systemic autoinflammatory diseases can classified according to several criteria. In this review, the underlying pathophysiological mechanism was adopted. Clinical features and the Mendelian inheritance of the different SAIDs described to date are summarized in Figure 3.

### 3.1. Cryopirin-Associated Periodic Syndrome (CAPS)

Cryopirin-associated periodic syndrome comprises a heterogeneous group of diseases caused by NLRP3 (nod-like protein family pyrin domain containing-3) GOF mutations encoded in chromosome 1q44 [16]. Autosomal dominant inheritance with multiple clinical phenotypes has been reported. However, somatic cases have also been described. NLRP3 encodes cryopyrin, which plays an essential role in IL-1β production. Hyperactivation of the inflammasome by NLRP3 GOF mutations induces IL-1β overproduction, leading to uncontrolled and inappropriate systemic inflammation [17,18]. The prevalence of CAPS is around three persons per million, and there is no gender or ethnic predilection [17].

This syndrome includes a continuum of three clinical phenotypes, from the milder familial cold autoinflammatory syndrome (FCAS) to the more severe Neonatal Onset Multisystem Inflammatory Disease (NOMID), also known as Chronic Infantile Neurologic Cutaneous Articular (CINCA) syndrome. Muckle-Wells syndrome (MWS) is an intermediate phenotype. Patients with FCAS present with self-limited (<24 h) episodes of fever, urticaria-like skin lesions, arthralgia, and conjunctivitis triggered by cold exposure. Muckle-Wells syndrome is clinically similar but has a chronic course and may progress to sensorineural deafness and AA amyloidosis (30% of cases), characterized by nephrotic syndrome and kidney failure [19]. The most severe presentation is NOMID/CINCA, an early-onset disease (usually before 6 months of age) that leads to death before adulthood unless controlled by timely intervention [20]. Affected infants fail to thrive and may develop bony overgrowth, joint contractures, destructive arthropathy, dysmorphism, learning disability, and progressive neurologic impairment [21]. Diagnostic criteria for CAPS consist of one positive inflammatory marker plus two or more typical symptoms: urticaria-like skin lesions (neutrophilic perivascular infiltrate), cold-induced episodes, sensorineural hearing loss (secondary to chronic cochlear inflammation), musculoskeletal symptoms (arthralgia/arthritis/myalgia), chronic aseptic meningitis and skeletal abnormalities such as epiphyseal overgrowth or frontal bossing [22,23].

In the largest CAPS cohort (n = 136) investigated to date, 40% of patients had neurological manifestations like headache (70%), papilledema (52%), hearing loss secondary to cochlear inflammation (42%), aseptic meningitis (26%), hydrocephalus (18%), mental retardation (16%) and seizures (4%) [24]. In the severe forms of the disease, permanent central nervous system (CNS) damage may occur in untreated patients, leading to brain atrophy, ventriculomegaly, arachnoid adhesions, and leptomeningeal enhancement.

Ancillary test results are usually non-specific but indicative of an inflammatory state. Chronic anemia and elevation of acute phase reactants during inflammatory episodes, including erythrocyte sedimentation rate (ESR), c-reactive protein (CRP), and serum amyloid A (SAA) protein, are the major laboratory findings [21]. Cerebrospinal fluid (CSF) analysis often reveals elevated intracranial pressure, pleocytosis, high protein levels, and normal glucose levels [23]. A predilection for high frequencies (4000–8000 Hz) and cochlear enhancement in inner ear MRI studies have been reported in patients with hearing loss.

Treatment with IL-1 inhibitors should be promptly started. Nonsteroidal (NSAIDs) and steroidal anti-inflammatory drugs can be prescribed to control symptoms but are not indicated as primary maintenance therapy [8]. Patients should be regularly monitored using complete blood count and ESR/CRP, disease activity scores, audiometry, ophthalmological examination, and urine protein test. Periodic cognitive assessment, lumbar puncture, brain MRI, and skeletal imaging are also indicated in severe cases [25,26].

### 3.2. Familial Mediterranean Fever (FMF)

Familial Mediterranean Fever is an autosomal recessive inherited disease caused by a mutation in the MEFV (Mediterranean FeVer) gene. This gene encodes pyrin and is located in chromosome 16 [26]. Heterozygous forms of FMF have recently been described and grouped as PAAND (Pyrin-Associated Autoinflammation with Neutrophilic Dermatosis). Pyrin overexpression induces constitutive inflammasome activation, leading to IL-1β and IL-18 overproduction [27,28,29].

Familial Mediterranean Fever is the most common hereditary AID and affects primarily eastern Mediterranean populations (especially non-Ashkenazi Jews, Armenians, Turks, and Arabs) [26]. The condition tends to manifest during childhood or adolescence and shows a slight male predilection (1.5–2 times) [26,30]. However, symptoms are first noted in adulthood in up to 10% of cases.

The disease is characterized by recurrent self-limiting attacks of fever or serositis and clinical manifestations such as abdominal or chest pain, arthritis, and erysipelas-like erythema [30,31]. Interestingly, polyarteritis nodosa and Henoch-Schoenlein purpura may also be associated with FMF [32]. Neurological symptoms may be present, the most common being aseptic meningitis, headache, demyelinating lesions, and pseudotumor cerebri [33].

Hypothetical associations between FMF and multiple sclerosis (MS) have been suggested since the physiopathology of both diseases involves increased IL-1β levels. According to a systematic literature review, MEFV heterozygosity is not associated with MS, but it is more common in FMF patients. Patients with MS and FMF are more likely to lack oligoclonal bands. Neurologists should suspect MS whenever neurological manifestations are detected in patients with FMF [34].

Familial Mediterranean Fever should be investigated in patients from the Eastern Mediterranean region presenting with recurrent inflammation. Eurofever Registry classification criteria may also be helpful [35,36]. The treatment of choice is lifelong colchicine [37,38]. Refractory cases may benefit from IL-1 or TNF inhibitors [37,38,39].

### 3.3. Mevalonate Kinase Deficiency (MKD) and Mevalonic Aciduria (MVA)

Mevalonate kinase deficiency and mevalonic aciduria (also known as hyper-IgD syndrome, or HIDS) are autosomal recessive diseases with similar genetic backgrounds to pathogenic MVK (mevalonate kinase) variants [40]. In both cases, clinical disease severity is inversely related to the remnant enzyme activity [41,42].

Mevalonate kinase deficiency is the most severe of the two conditions and manifests as failure to thrive, stillbirth, or congenital malformations such as shortened limbs and dysmorphic craniofacial features [43,44]. Affected infants may also develop hypotonia, psychomotor retardation, cataract, and myopathy [43]. The disease is characterized by recurrent fever attacks, which tend to occur every 2–8 weeks and last 3–7 days. Other clinical manifestations include vomiting and diarrhea, arthritis/arthralgia, cervical lymphadenopathy, subcutaneous edema, and rash [43]. Hepatosplenomegaly may occur in the chronic stage of the disease or during episodes of fever [42,43].

The MKD/HIDS clinical phenotype is milder and marked by recurrent febrile episodes, which start at around 6 months of age and are often triggered by vaccination. These episodes last around 4 days and recur at irregular intervals. Attacks are characterized by painful cervical lymphadenopathy, abdominal pain, vomiting, diarrhea, aphthous ulcers, arthralgia, myalgia, and fatigue [42,43]. Neurological symptoms consist primarily of headaches during disease flares. Mental retardation, cerebellar syndrome, and aseptic meningitis have been reported in a minority of patients [43].

The diagnosis of MVA and MKD is based on increased urinary mevalonic acid levels, which is considered a pathognomonic finding [44,45]. Although not limited to and not always seen in MKD, immunoglobulin D levels may also be elevated [45]. Measurement of MVK activity in fibroblasts and lymphoblasts (close to zero in MVA and 1–10% of normal levels in MKD) is another useful diagnostic test [41]. The diagnosis of MKD/MVA is confirmed by genetic identification of MVK pathogenic variants [44,45].

Mevalonate kinase deficiency therapy is usually supportive. However, hematopoietic stem cell transplant can be used to control fever episodes and inflammation in patients with severe disease [46]. Treatment with NSAIDs or steroids can be prescribed to relieve acute symptoms. Anakinra is also indicated during disease flares. Maintenance therapy with canakinumab, anakinra, or TNF-blockers (etanercept or adalimumab) should be considered in cases with persistent or frequent systemic inflammation [47,48].

### 3.4. Type I Interferonopathies

Interferons (IFN) are a family of cytokines with a pivotal role in viral infection control, cell multiplication, and immune responses. Three types of IFNs have been described: type I IFN (particularly IFN-α, β, ε, κ, and ω), type II (or IFN-γ) and type III (or IFN-λ) [49,50]. Type I IFNs are secreted by macrophages, lymphocytes, dendritic cells, fibroblasts, and hematopoietic plasmacytoid dendritic cells [51,52]. Type I IFN-mediated responses often occur after the recognition of free cytosolic nucleic acids by intracellular receptors. Therefore, abnormal signaling induced by defective DNA/RNA clearance typically promotes a group of diseases characterized by chronic hyperactivation of the IFN pathway and the production of large amounts of type I IFN [51,52], the so-called type I interferonopathies. The classical phenotype is Aicardi-Goutières syndrome [53,54].

Despite clinical heterogeneity, some phenotypic features, such as early skin vasculopathy with chilblains, livedo reticularis, panniculitis, CNS involvement, and interstitial lung disease, are shared by different interferonopathies [50,51,52].

Neurological involvement is a prominent feature of most interferonopathies and probably reflects the role of type I IFN in microglial function regulation [54]. Under normal circumstances, type I IFN stimulates microglial phagocytosis and helps maintain the integrity of the blood–brain barrier. Interferons have dual and opposite effects on the CNS and microglia. Murine MS and stroke models have shown that IFNβ produced by microglia modulates myelin sheath inflammation via increased phagocytosis and safeguards blood-brain barrier integrity. Chronic upregulation of type I IFN signaling in microglia is harmful, particularly in the white matter [55]. The fetal CNS is more susceptible to IFN-related insults due to developmental processes such as neurogenesis, synaptogenesis, synaptic pruning, and myelination [54,55]. Early onset basal ganglia calcifications, epilepsy, and psychomotor retardation may occur in some interferonopathies which resemble congenital TORCH infections: Toxoplasmosis, Other (syphilis, varicella-zoster, parvovirus B19), Rubella, Cytomegalovirus and Herpes congenital infections (pseudo-TORCH syndrome) [56].

Aicardi-Goutières syndrome (AGS) is the prototypical type I interferonopathy. However, other important diseases have similar pathophysiology, including spondyloenchondrodysplasia (SPENCD), monogenic forms of systemic lupus erythematosus (SLE), proteasome-associated autoinflammatory syndromes (PRAAS), ISG15 (interferon-stimulated gene 15) deficiency, Singleton–Merten syndrome (SMS), COPA (coatomer protein subunit alpha) syndrome, STING-associated vasculopathy with onset in infancy (SAVI), and Retinal vasculopathy with cerebral leukoencephalopathy and systemic manifestations (RVCL-S) [52,53,54].

Aicardi-Goutières syndrome is a clinically heterogeneous disease with two major phenotypes: early-onset and late-onset. Early-onset disease is characterized by psychomotor developmental delay and neonatal liver abnormalities, whereas late-onset disease manifests as progressive head growth decline, spasticity, and moderate to severe global developmental delay after normal development in the first weeks or months of life. Abnormal eye movements, glaucoma, visual defects, and startle reactions to sensory stimuli are other common signs [55].

Autosomal recessive inheritance has been described in homozygous or compound heterozygous mutations such as TREX1, RNASEH2C, RNASEH2A, RNASEH2B, SAMHD1 (SAM and HD domain-containing deoxynucleoside triphosphate triphosphohydrolase), and ADAR1 (adenosine deaminase acting on RNA 1). Autosomal dominant inheritance is primarily caused by IFIH1 (IFN-induced helicase C domain-containing protein 1) heterozygous variants. Stroke and cerebral aneurysms are common in SAMHD1-deficient patients, whilst bilateral striatal necrosis is often seen in patients with AGS-associated mutation in ADAR1 [56,57,58].

Retinal vasculopathy with cerebral leukoencephalopathy and systemic manifestations is a rare autosomal dominant vasculopathy caused by mutations in the TREX1 gene. Interestingly, this gene may also be associated with an AGS phenotype. Therefore, TREX1-associated RVCL-S is a good example of the heterogeneous nature of type I interferonopathies. The mean age at disease onset is usually 30–50 years. Clinical signs include progressive blindness, focal neurological signs, vascular dementia, and systemic manifestations such as Raynaud’s phenomenon, anemia, and liver and kidney failure [59]. Focal neurological symptoms and cognitive impairment have been described in 68% and 56% of symptomatic patients, respectively [60]. Psychiatric symptoms (e.g., bradyphrenia, apathy, and irritability) are the major complaints, while migraines (with or without aura) and seizures are less common [61]. Computed tomography and MRI abnormalities described so far are limited to the white matter. T2 hyperintense lesions with long-lasting enhancement, particularly in the periventricular and deep white matter, are the typical findings [62,63,64].

Despite variable clinical manifestations, the diagnosis of type 1 interferonopathies can be confirmed by a typical “interferon signature” (i.e., IFN-induced gene expression upregulation and IFN-inhibited gene expression downregulation). Expression of more than 20 genes whose transcription is primarily regulated by type 1 interferon signaling can be detected using multiplex quantitative polymerase chain reaction (qPCR). This method has been clinically validated and is more accurate than isolated measurement of serum IFN levels. Serum IFN measurement involves significant preanalytical challenges (low stability, rapid analyte degradation, and large coefficients of variation) and may not reflect actual cytokine levels. Therefore, it is not often recommended [65]

The primary transcription factor in type I interferon signaling is STAT1-dependent. Hence, pharmacological treatment with Janus kinase 1 and 2 inhibitors (JAKis), particularly baricitinib, is widely used in patients with type I interferonopathies. Ruxolitinib (JAK 1 and 2) and tofacitinib (JAK 1, 2, and 3) are other potential therapeutic alternatives. This treatment modality decreases the expression of interferon signaling genes and alleviates AGS-related symptoms, including neurological manifestations, fever, and skin inflammation [65,66].

### 3.5. Tumor Necrosis Factor Associated Periodic Syndrome (TRAPS)

Tumor Necrosis Factor Associated Periodic Syndrome is an autosomal dominant disease caused by mutations in TNFRSF1A (tumor necrosis factor receptor superfamily 1A) gene, which encodes TNF receptor 1 [67]. The pathophysiology of this syndrome remains unclear. However, associations with aberrant protein folding and resultant persistent inflammation have been suggested [68]. The disease is characterized by recurrent episodes of myalgia, prolonged fever (usually lasting 1–3 weeks), migratory rashes, headache, serositis, arthralgia, abdominal pain, and periorbital edema. Attacks may be spontaneous or triggered by factors like stress or infection, with monthly or yearly recurrence patterns [68]. Systemic amyloid A (AA) amyloidosis occurs in more than 10% of affected patients and is associated with a poor prognosis [69].

Neurological symptoms are uncommon. Still, around 20% of patients experience headache. Seizures (1%), vertigo (1%), diplopia, and cerebrovascular lesions have also been reported [70]. In some cases, ancillary tests may reveal CSF pleocytosis and white matter lesions [71]. As in other SAIDs, TRAPS diagnosis is confirmed by genetic testing. TNF inhibitors are the gold standard treatment. However, IL-1 blockers can be used in refractory cases [72].

### 3.6. A20 Haploinsufficiency (HA20)

Protein A20 is encoded by the TNFAIP3 gene and plays a key role in the modulation of the NF-kB canonical pathway. This protein targets the inhibitor of nuclear factor kappa B kinase subunit gamma IKKγ (also known as NEMO and an NF-kB essential modulator) and the receptor-interacting protein kinase 1 (RIPK1) [73]. Haploinsufficiency of A20 is caused by high penetrance heterozygous LOF germline mutations in the TNFAIP3 gene. Despite the earlier onset of symptoms, affected patients often meet clinical diagnostic criteria for Behçet’s disease (BD). HA20 is thought to be a monogenic form of BD [74,75], and sporadic BD is the first diagnosis in more than 70% of cases.

The vast majority of patients with HA20 present with recurrent painful oral, genital, and/or gastrointestinal ulcers. Other common symptoms are gastrointestinal complaints, including bloody diarrhea, polyarthritis and/or arthralgia, skin lesions (pseudofolliculitis, acne, and dermal abscesses), and ocular findings such as anterior uveitis and retinal vasculitis. Neurological manifestations are uncommon. However, a few cases of CNS vasculitis have been described [76].

Acute-phase reactants tend to be particularly elevated during relapses and normal in intercritical periods. Fluctuating low autoantibody titers have been reported, including antinuclear antibodies, anti-dsDNA, anti-Sm/RNP, lupus anticoagulant, and anticardiolipin antibodies [75,76]. Pathergy phenomenon and HLA-B51 may also occur in some patients. Although BD-like symptoms are present in approximately 75% of HA20 patients, cases with SLE, juvenile idiopathic arthritis, psoriatic arthritis, and even PFAPA features have recently been described, suggesting a broader clinical spectrum. As in other SAIDs, the diagnosis can be confirmed by gene sequencing.

Anticytokine agents such as anti-TNF or anti-IL-1 are effective in suppressing systemic inflammation in most cases. Hematopoietic stem cell transplant may be indicated in patients with severe refractory disease [73,74].

### 3.7. Blau Syndrome (BS)

Blau syndrome (BS), also known as early-onset sarcoidosis, is a hereditary autosomal dominant disorder resulting from a mutation in the NOD2 (nucleotide-binding oligomerization domain protein 2, or CARD15, caspase recruitment domain family 15) gene. NOD2 is an intracytoplasmic receptor involved in the innate immune response to bacteria [75,76,77]. Upon activation, it initiates the formation of granulomatous lesions by triggering the activation of NF-kB.

The classical triad of BS includes polyarthritis, uveitis, and rash; however, visceral and vascular manifestations have been observed in 29–48% of BS patients, underscoring its systemic nature. Cutaneous manifestations are predominant in BS [75,76]. In early infancy, patients often develop asymptomatic dermatitis characterized by a tan-colored rash and an ichthyosis-like exanthema. Panniculitis resembling erythema nodosum and leukocytoclastic vasculitis can also occur. Histopathologically, the findings are akin to those seen in sarcoidosis, featuring noncaseating granulomas and giant nucleated cells that stain positive with periodic acid-Schiff staining [77,78].

Joint involvement represents the most common noncutaneous manifestation and typically initiates in the first decade, frequently leading to camptodactyly. Ocular involvement is a hallmark of BS, with chronic granulomatous anterior uveitis being the most common ocular manifestation. This presents with ocular redness, pain, photophobia, and blurred vision and can lead to complications such as cataracts, glaucoma, and visual loss. Bilateral panuveitis is frequently observed [75,76,77].

Less commonly, the granulomatous infiltrate can affect organs such as the liver, lungs, kidneys, nervous system, and arteries [75,77]. Nervous system involvement was first described by Jabs et al. in 1985, with reported cases including corticosteroid-responsive bilateral neurosensory hearing loss and transient sixth nerve palsy [78]. Emaminia et al. reported a case of seizures in one of three family members with BS [79]. Optic nerve involvement with papilledema and optic neuropathy has also been documented [80].

Diagnosis relies on clinical criteria, which include an onset before the age of 5, the presence of classical symptoms (dermatitis, arthritis, and uveitis), and the identification of noncaseating granulomas. Additionally, confirmation using DNA sequencing is required. However, the NOD2 gene exhibits limited conservation across different species, and polymorphisms are common, which can make sequencing challenging [76,77,78]. Treatment options typically include steroids, anti-TNF agents, and immunosuppressants. Relapses often occur upon treatment discontinuation [78].

### 3.8. Deficiency of Adenosine Deaminase 2 (DADA2)

DADA2 is a monogenic autoinflammatory disease caused by loss-of-function (LOF) mutations in the CECR1 (cat eye syndrome chromosome region, candidate 1) gene. This gene encodes ADA2, which is highly expressed in immune cells, particularly those of the myeloid lineage. ADA2 plays a crucial role in hematopoietic cell maturation and the maintenance of vascular integrity [4,79]. In DADA2, there is a predominant polarization of M1 (pro-inflammatory) over M2 (anti-inflammatory) macrophages, creating an inflammatory environment characterized by pericyte dysfunction in blood vessel walls. This dysfunction ultimately leads to vasculitis and ischemia. The pathophysiology of DADA2 involves both the TNF and IFN pathways [81,82].

DADA2 is characterized by systemic vasculitis, early-onset stroke, bone marrow failure, and immunodeficiency. The broad clinical spectrum of this condition can be divided into four phenotypes: inflammatory and/or vasculitic, hematologic, immunodeficient, or presymptomatic. The phenotypes, except for the presymptomatic one, can overlap. Uncommon presentations of DADA2, such as Castleman disease, antiphospholipid syndrome, and lymphoproliferative disease, can also occur. Patients with the inflammatory phenotype are particularly important for neurologists, as they may manifest with cutaneous vasculitis (livedo racemose), ischemic stroke, intracranial hemorrhage, and neuropathy and are often misdiagnosed with polyarteritis nodosa or Sneddon syndrome [81,82,83].

The diagnosis is based on clinical criteria, including measurement of plasma or serum ADA2 enzymatic activity or sequencing of the ADA2 gene. The initial evaluation should encompass a complete laboratory study (including serum immunoglobulin levels and lymphocyte subsets), brain magnetic resonance imaging and angiography, abdominal ultrasound with Doppler, and electrocardiogram. Additional testing should be considered based on specific manifestations [83].

Tumor necrosis factor inhibitors (TNFi) are the treatment of choice for patients with the inflammatory/vasculitic phenotype, significantly reducing the risk of strokes and other vasculitic organ injuries. Soluble TNF receptor (etanercept) and monoclonal antibodies against TNF (adalimumab, infliximab, and golimumab) all appear to be effective. If bone marrow failure and/or immunodeficiency are also present, the risk of infection should be considered alongside chronic immunosuppression. To minimize the development of neutralizing antibodies, concurrent use of methotrexate should be considered [82,83].

In the event of acute stroke in a patient with DADA2, neuroprotective strategies and anti-inflammatory treatment with glucocorticoids and/or TNFi should be initiated. The use of antiplatelet, anticoagulant, and antithrombotic therapy is controversial, as these agents pose a risk of secondary hemorrhagic stroke conversion and are therefore not recommended. Hematologic features of DADA2 are refractory to immunomodulatory therapy, and allogeneic hematopoietic stem cell transplantation (HSCT) is a curative option in such cases [83].

## 4. General Approach to SAIDs

### General Approach

Systemic autoinflammatory diseases are uncommon and often underdiagnosed conditions with heterogeneous clinical manifestations. Treatment is, therefore, challenging. Given the crucial role of neurological signs in decision-making regarding immunosuppression [4,82,83,84], general neurologists must be aware of these disorders. Recurrent flares and subclinical manifestations are common in SAIDs, particularly in CAPS. Hence, special attention should be given to regular monitoring of patients with neurological impairments and hearing loss. Assessment by a multidisciplinary team, including neurologists, rheumatologists, immunologists, physiotherapists, and psychologists, is recommended. Treat-to-target (T2T) should be aimed at complete remission or minimal disease activity [83,84].

Since specific biomarkers are lacking, comprehensive physical examination combined with disease activity and damage index scores is essential to monitor disease activity and prevent potentially irreversible end-organ damage. The autoinflammatory diseases activity index (AIDAI) is a validated tool for the clinical assessment of FMF, cryopirinopathies, TRAPS, and MVK. Disease activity should be self-reported daily. A monthly cutoff score of 9 is then used to distinguish between active and inactive disease (Figure 4) [85].

The autoinflammatory damage-disease-damage index (ADDI) can be used to assess irreversible damage in cases of SAIDs. This questionnaire comprises eight domains. The total score ranges from 0 to 27, and no cutoff has been determined. The higher the score, the greater the cumulative damage caused by a given AID [85,86]. Routine laboratory tests tailored to specific AIDs are also recommended. Measurement of acute phase reactants such as ESR, CRP, and SAA during disease flares and intercritical periods may be helpful. Of note, SAA is also an important diagnostic marker for systemic amyloidosis [3,10]. Target-organ amyloidosis should be confirmed by other specific tests, such as kidney biopsy and cardiac magnetic resonance imaging.

Infections are the most important differential diagnosis in patients with AIDs. Inflammatory markers alone may not be enough to discriminate between these conditions. Therefore, procalcitonin measurement and bacterial or fungal cultures are recommended for accurate diagnosis [10].

Inactivated vaccines are thought to be safe and should be indicated according to country-specific vaccination guidelines. Live vaccines should be avoided in patients undergoing immunosuppressive therapy [3,10].

Table 1 presents an overview of the pathophysiology of the diseases discussed in the article, along with the proposed treatment strategies.

## 5. Conclusions

Systemic autoinflammatory diseases are potentially life-threatening conditions. Early diagnosis offers a unique opportunity for appropriate treatment and prevention of permanent damage, with significant contributions to the patient’s quality of life. Neurological symptoms have been reported in most SAIDs. Therefore, general neurologists must be able to recognize these conditions, which should be part of the list of possible differential diagnoses, particularly in patients with recurrent unprovoked inflammatory attacks. Once the diagnosis has been confirmed, assessment by a multidisciplinary team, including immunologists or rheumatologists, is strongly recommended for timely discussion of therapeutic alternatives.

## Figures and Tables

**Figure 1 brainsci-13-01351-f001:**
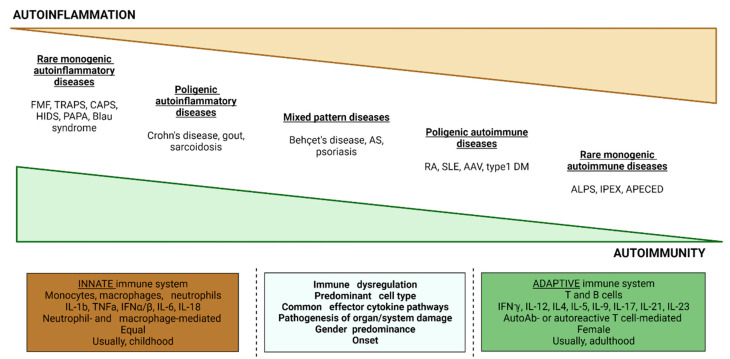
Comparison and intersection between autoinflammation and autoimmunity concepts. Abbreviations: AAV, ANCA-associated vasculitis; APECED, autoimmune polyendocrinopathy, candidiasis ectodermal dystrophy; AS, ankylosing spondylitis; ALPS, autoimmune lymphoproliferative syndrome; CAPS, cryopyrin-associated periodic syndromes; DM, diabetes mellitus; FMF, familial Mediterranean fever; HIDS, hyperimmunoglobulinemia D and periodic fever syndrome; IPEX, immune dysregulation, polyendocrinopathy, enteropathy, X-linked; PAPA, pyogenic arthritis, pyoderma gangrenosum, and acne syndrome; RA, rheumatoid arthritis; SLE, systemic lupus erythematosus; TRAPS, tumor necrosis factor associated periodic syndrome. Adapted from [8,9].

**Figure 2 brainsci-13-01351-f002:**
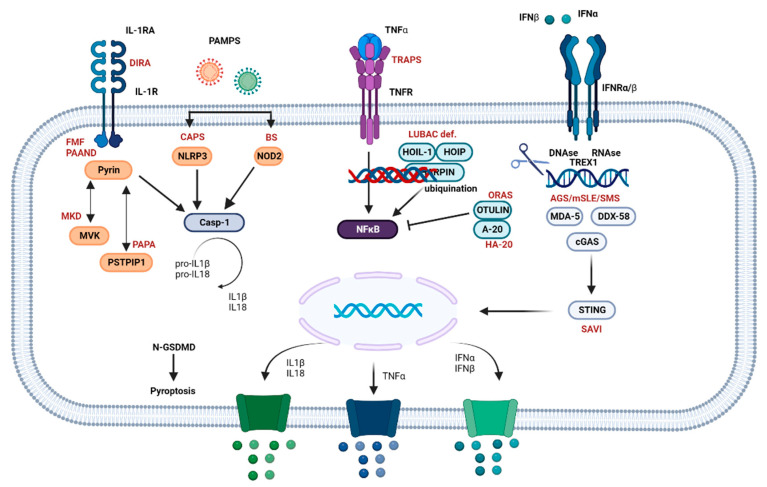
Inflammatory downstream pathways of autoinflammatory diseases (AIDs). The inflammasome pathway is activated by various pathogen-associated molecular patterns (PAMPs), initiating a cascade that culminates with the activation of pyrin or nucleotide-binding domain, leucine-rich–containing family, pyrin domain–containing-3 (NLRP3) or nucleotide-binding oligomerization domain 2 (NOD2) towards different intracellular sensors, such as pattern recognition receptors (PPRs). Mevalonate kinase (MVK) and proline-serine-threonine phosphatase interacting protein 1 (PSTPIP1) modulate the pyrin pathway. Inflammasome receptors lead to caspase 1 activation, which converts pro-IL-1 and pro-IL-18 into their bioactive forms, such as IL-1β and IL-18. Caspase 1 also cleaves gasdermin-D (GSDMD), with its N-N-terminal domain (GSDMD-N) forming cytotoxic pores in the cellular membrane, resulting in pyroptosis and cytokine release. The canonical nuclear factor kappa B (NF-kB) activation pathway is triggered by the interaction between tumor necrosis factor-alpha (TNFα) and the TNF receptor (TNFR), followed by the phosphorylation of linear ubiquitin assembly complex (LUBAC), primarily composed of HOIP, HOIL-1, and SHARPIN, which is necessary for efficient activation. A20 and OTULIN modulate the pathway by cleaving and activating polyubiquitin. The interferon (IFN) pathway is activated by the interaction between IFNα, IFNβ, and the IFNα/β receptor (IFNR), leading to the activation of deoxyribonucleases (DNAse), ribonucleases (RNAse), three prime repair exonuclease 1 (TREX1), cyclic GMP-AMP synthase (cGAS), melanoma differentiation-associated protein 5 (MDA5), and DEAD Box Protein 58 (DDX58), which activate stimulator of interferon genes (STING). The latter translocates to the nucleus, stimulating the transcription of type I IFN genes. STING also directly activates NFkB signaling, resulting in the release of IL-1, TNFα, IFNα, and IFNβ. Some diseases are shown in the figure to illustrate potential immune dysregulations that may occur. Abbreviations: AGS, Aicardi-Goutières syndrome; CAPS, cryopyrin-associated periodic syndrome; DIRA, deficiency of interleukin-1 receptor antagonist; FMF, familial Mediterranean fever; HA-20, haploinsufficiency A20; LUBAC deficiency, linear ubiquitin chain assembly complex deficiency; MKD, mevalonate kinase deficiency; mSLE, monogenic systemic lupus erythematosus; ORAS, OTULIN-related autoinflammatory syndrome; PAAND, pyrin-associated autoinflammation with neutrophilic dermatosis; PAPA, pyogenic arthritis, pyoderma gangrenosum, and acne; SAVI, STING-associated vasculopathy with onset in infancy; SMS, Singleton-Merten syndrome; TRAPS, tumor necrosis factor receptor-associated periodic syndrome. Adapted from Donato et al. [15]. Created using BioRender.

**Figure 3 brainsci-13-01351-f003:**
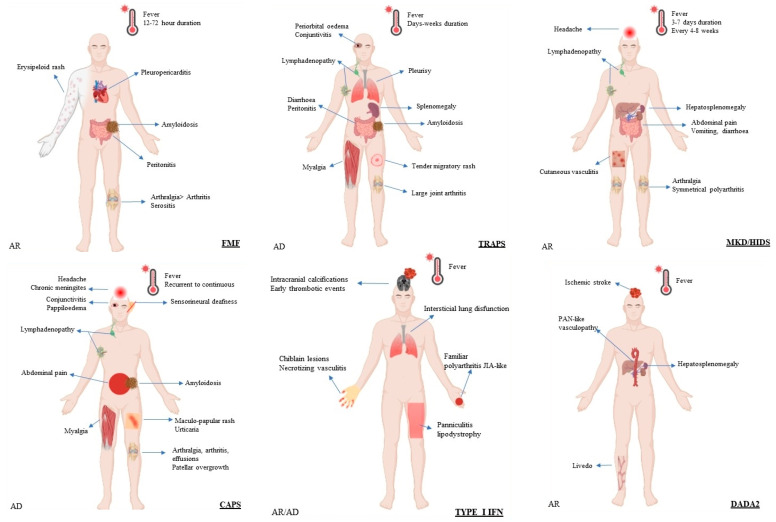
Schematic representation of selected systemic autoinflammatory disorders in adults. Abbreviations: AD, autosomal dominant; AR, autosomal recessive; BS, Blau syndrome; CAPS, cryopyrin-associated periodic syndrome; DADA2, deficiency of adenosine deaminase 2; HIDS, hyperimmunoglobulin D syndrome; FMF, familial Mediterranean fever; IBD, inflammatory bowel disease; JIA, juvenile idiopathic arthritis; MKD, mevalonate kinase deficiency; PAN, polyarteritis nodosa; TRAPS, TNF-receptor associated periodic syndrome.

**Figure 4 brainsci-13-01351-f004:**
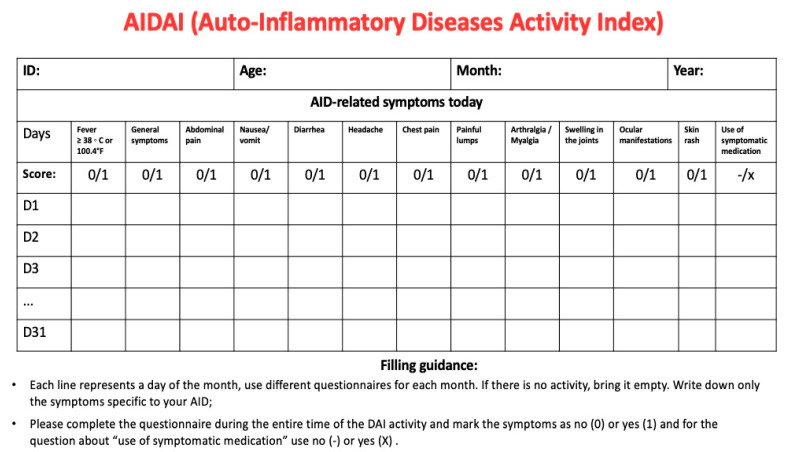
Autoinflammatory diseases activity index (AIDAI). The questionnaire is exclusively validated for familial Mediterranean fever, cryopyrinopathies, tumor necrosis factor-associated periodic syndrome, and mevalonate-kinase deficiency patients. All information must be daily completed by the patient or guardians (ex. parents) according to 13 domains: (a) fever ≥38 °C (100.4 °F); (b) general symptoms; (c) abdominal pain; (d) nausea/vomiting; (e) diarrhea; (f) headaches; (g) chest pain; (h) painful nodules; (i) arthralgia or myalgia; (j) swelling of the joints; (k) ocular manifestations; (l) rash; (m) pain relief after use of analgesics. Answers are dichotomized as: no (0) = absence of symptom or yes (1) = presence of symptom. The total daily score ranges from 0–12, and monthly, from 0–372. An AIDAI monthly total score < 9 separates inactive patients from active AIDAI subjects (total score ≥ 9). Adapted from [85,86].

**Table 1 brainsci-13-01351-t001:** Autoinflammatory diseases are grouped according to pathophysiological features and proposed treatment.

Treatments		
IL-1β-Mediated Autoinflammatory Disorders	Cryopyrin-Associated Periodic Syndrome (CAPS)	IL-1 antagonists, steroids
	Familial Mediterranean Fever (FMF)	Colchicine, steroids, TNF antagonists, IL-6 antagonists, and IL-1 antagonists
	Mevalonate kinase deficiency (MKD) and mevalonic aciduria (MVA)	IL-1 antagonists, steroids, colchicine, IL-6 antagonists, and TNF antagonists
Relopathies	A20 Haploinsufficiency	anti-TNF, anti-IL-1, and hematopoietic stem cell transplant (severe and refractory disease)
Dysregulation of TNF activity	Blau syndrome	Steroids, TNF antagonist
	Deficiency of adenosine deaminase 2 (DADA2)	anti-TNF, and hematopoietic stem cell transplant
Type I interferonopathies	Aicardi-Goutières syndrome Proteasome-associated autoinflammatory syndromes (PRAAS) ISG15 (interferon-stimulated gene 15) deficiency Singleton–Merten syndrome (SMS) COPA (coatomer protein subunit alpha) syndrome STING-associated vasculopathy with onset in infancy (SAVI)	JAK inhibitors

## Data Availability

Not applicable.

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
