# Peer review of "What General Neurologists Should Know about Autoinflammatory Syndromes?"

_brainsci, 2023, doi:10.3390/brainsci13091351_

Round 1
Reviewer 1 Report
In this manuscript, Barsottini and colleagues discuss clinical and immunopathologic concepts of systemic autoinflammatory diseases (SAIDs), focusing particularly on neurologic manifestations. Overall, this paper is an interesting review. However, I still have some concerns about the current form of the manuscript. The authors need to address several aspects as follows:
Main concerns:
1. This review is a reference for neurologists, so the authors need to indicate the symptoms of Blau syndrome (BS) and deficiency of adenosine deaminase 2 (DADA-2) in systemic autoinflammatory diseases in the nervous system.
2. The authors have provided corresponding treatments in the classification of systemic autoinflammatory diseases, so why is there a separate listing of treatments later? And there is duplication of content, so the two sections should be combined.
3. Figure A3 takes up two pages and should be merged into one.
Minor concern:
Duplicate numbers labeled in front of the reference need to be removed from one of them.
Minor editing of English language required
Reviewer 2 Report
The paper presented to me for review is a very well-written, wide-ranging review article. The authors have clearly presented the autoinflammatory syndromes can present with various neurological manifestations. The paper is written in good language and is based on new literature. The topic is original or relevant in the field. In my opinion, it brings an important summary of knowledge in the field of autoinflammatory syndromes.
The only addition I would suggest is to add 2-3 sentences on the most common autoimmune disease that can manifest neurologically i.e. autoimmune thyroid disease based on: PMID: 36139446
Reviewer 3 Report
Dear Dr. Moraes,
Thank you for your great paper. I believe that it is a well-written review article. I just recommend to be fixed a few minor errors:
1. Please check the explanation of the figures because there are a few abbreviations that were not used in the figure.
2. Please fix the issue of twice-numbered references.
Thank you so much.
Best wishes
It is a well-written article.
